# Depth-First Proof-Number Search with Heuristic Edge Cost and Application to Chemical Synthesis Planning

**Akihiro Kishimoto**
IBM Research, Ireland

**Beat Buesser**
IBM Research, Ireland

**Bei Chen**
IBM Research, Ireland

**Adi Botea**
Eaton, Ireland*

## Abstract

Search techniques, such as Monte Carlo Tree Search (MCTS) and Proof-Number Search (PNS), are effective in playing and solving games. However, the understanding of their performance in industrial applications is still limited. We investigate MCTS and Depth-First Proof-Number (DFPN) Search, a PNS variant, in the domain of Retrosynthetic Analysis (RA). We find that DFPN's strengths, that justify its success in games, have limited value in RA, and that an enhanced MCTS variant by Segler et al. significantly outperforms DFPN. We address this disadvantage of DFPN in RA with a novel approach to combine DFPN with Heuristic Edge Initialization. Our new search algorithm DFPN-E outperforms the enhanced MCTS in search time by a factor of 3 on average, with comparable success rates.

## 1   Introduction

Search is a core AI technique, especially in games [5, 27, 31] and domain-independent planning [3, 12]. Historically, new search algorithms and novel combinations of existing algorithms have led to significant performance improvements. Another way to achieve progress is to transfer successful approaches from one domain to another. This paper combines both strategies.

Proof-Number Search (PNS) [1] and Monte Carlo Tree Search (MCTS) [18] are two notable algorithms that have been extensively studied in solving games/game positions and playing games, respectively. Their success has been demonstrated by solving the game of checkers [27] and by achieving super-human strength in playing the game of Go [31].

Due to recent advances in AI, chemistry is being reconsidered as an AI research domain [14]. We focus on chemical synthesis planning, the task of planning chemical reaction routes to synthesize a given organic molecule. Retrosynthetic Analysis (RA) is a technique going backwards from the target molecule towards a library of usually smaller, starting molecules. This task can be modeled similarly to solving games [11] and can be tackled with search algorithms such as PNS and MCTS [11, 29]. However, since no direct performance comparison between PNS and MCTS has been made, their search performance in RA is not well understood.

We investigate the search performance of MCTS and Depth-First Proof-Number (DFPN) Search [21], a variant of PNS, in RA. PNS variants [1, 21] are designed to efficiently identify the smallest amount of work needed to solve a game position. The game research community largely agrees that this advantage is one reason why PNS variants have been a popular choice in solving difficult games or game positions [16, 21, 27], with solution lengths exceeding 1500 steps. However, we discover that DFPN performs poorly in the domain of RA. We find that DFPN's significant performance degradation is due to RA's *lopsided search space*, which alternates large branching factors with very small ones. To address this, we introduce DFPN-E, a novel algorithm to combine DFPN with

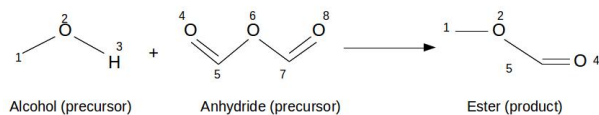

Figure 1: Example of reaction rule. Esterification reaction adapted from [11].

Heuristic Edge Initialization. We empirically evaluate DFPN-E and an enhanced MCTS variant [29] on a dataset of complex target molecules extracted from the US patent literature. We demonstrate that DFPN-E outperforms this enhanced MCTS in search time, with competitive success rates of finding successful synthesis routes.

## 2  Chemical Synthesis Planning

Chemical synthesis planning is the task of creating a sequence of chemical reactions (or reaction pathway) that synthesizes a target organic molecule by using existing (e.g., commercially available) starting materials. Chemists perform *retrosynthetic analysis* (RA), a systematic analysis that recursively splits a target molecule into simpler reactant molecules with reactions they expect to be feasible, until they find a branching reverse route from the target to starting materials. Automated retrosynthetic analysis has been an open problem for 50 years, e.g., [6, 7, 33], because of its scientific importance in chemistry as well as its direct applications to pharmaceutical and material industries.

RA takes as input a target molecule, a library of starting materials and a database of reactions. A molecule is represented as a graph where nodes and links correspond to atoms and chemical bonds, respectively. A starting material database contains definitions of available starting molecules. A reaction database has a set of *reaction rules* specifying graph substructures of molecules that must be present to allow the application of the reaction. Reaction rules can be constructed manually [11, 33] or automatically [19, 29].

Since RA explores the space backwards, from the target molecule, reaction rules are applied in a reverse (or retrosynthetic) manner. When checking if a reaction rule can be applied to a target molecule, RA checks if the target has a substructure specified in the product of the rule. If this is the case, the target is split into the reactant molecules specified in the reaction rule. Unless a reactant molecule $P$ appears in the starting material database, $P$ needs to be synthesized, recursively applying a reaction rule to $P$. A molecule may have multiple matching substructures where a reaction rule can be applied. In this paper, all these are considered as different moves.

Figure 1 illustrates a reaction rule for producing an ester from an alcohol and anhydride. The carbon atoms located in numbers 1, 5 and 7 are omitted. To activate this reaction, an alcohol and anhydride molecules need to have substructures as illustrated in the figure. An ester is produced by modifying the substructures specified there and keeping the remaining substructures unchanged.

Our focus is to elucidate the behavior of MCTS and PNS in the search space of RA, modeled in [11], where each reaction rule captures only substructural modifications between a product and its reactants. In practice, reaction rules need to consider constraints such as protecting groups, side compounds, temperature and yield [33]. This is beyond the scope of our paper.

While RA is often modeled as single-agent search (OR search), e.g., [19, 29, 33], Heifets and Jurisica model RA as a problem similar to solving a game position in two-player games such as chess and Go [11]. Their model allows to efficiently split a problem into a set of independent subproblems that can be solved with AND/OR search. In their game position solving framework, two players alternately play moves. Terminologies of wins and losses are used from a viewpoint of the first player. The first player attempts to synthesize a target molecule, while the second player (or opponent) attempts to prevent it. A position of the first player corresponds to a molecule they want to synthesize. A move of the first player is a reaction rule applicable to that molecule in a reverse manner. On the other hand, a position for the opponent corresponds to a set of precursor molecules generated by a reaction rule. A move for the opponent is to select one of the precursor molecules in their position. In Figure 1, an application of this rule corresponds to a move of the first player if it is applicable. The opponent has two moves, one choosing an alcohol and the other choosing an anhydride.

A molecule in the starting material database is a winning terminal position, while a molecule to no applicable reaction rule is a loss terminal position. At a position for the first player, if at least one move leads to a win, that position is a win, as one reaction to synthesize the molecule exists. If all legal moves lead to losses, the position is a loss (i.e., no pathway exists to synthesize it). At a position for the opponent, if all moves (precursor molecules presented to the first player) lead to wins, the position is a win (all precursors are successfully generated). Otherwise, the position is a loss.

The size of the RA search space can be very large. The branching factor at OR nodes is typically estimated as 80 and a synthetic route may require up several tens of synthetic steps [33]. In addition, complex molecules may require over 100 steps [11]. Like in many games, the search space of RA is cyclic since a sequence of reaction rules may create repeated states, e.g., oxidation and reduction, while a solution may not contain a cycle (see the next section for the definition of a solution). Unlike in many games, successor state generation is a serious bottleneck in RA, because checking an applicability of a reaction rule requires to solve subgraph matching, which is an NP-complete problem. For example, Kishimoto et al. [14] discuss that the implementation in [11] generates at most several tens of molecules per second, as compared to millions of positions per second in a game such as chess.

## 3 Background

This section gives an overview of PNS and MCTS. See [4, 17] for more comprehensive surveys.

### 3.1 AND/OR Search and Proof-Number Search Variants

In our model, OR and AND nodes correspond to the first player and the opponent, respectively. OR and AND nodes alternate along a given pathway in the search tree. A "move" represents an edge (transition) in a search tree. Once the AND/OR search finds a solution, the value of the root node is determined to be either a win or a loss. Node $n$ is called *proven* if the value of $n$ is proven to be a win that is also called a *proof*. Node $n$ is called *disproven* if the value of $n$ is proven to be a loss. A *disproof* is also used to express a loss. The value of $n$ is *unknown* or *unproven* if $n$ is neither proven nor disproven.

A *proof tree* $T$ of node $r$ theoretically ensures that $r$ is proven. $T$ has the following properties: (1) The root node $r$ is in $T$, (2) for each internal OR node $n$ in $T$, at least one child of $n$ is in $T$, (3) for each internal AND node $n$ in $T$, all the children of $n$ are in $T$, and (4) all terminal nodes in $T$ have values of wins. A *disproof tree*, that provides a disproof, is analogously defined.

An efficient AND/OR search aims at finding a proof or disproof tree as quickly as possible. PNS uses proof and disproof numbers to estimate the difficulty of finding a proof or a disproof for node $n$ [1]. The proof number $pn(n)$ for node $n$ is defined as the minimum number of leaf nodes to be proven to find a proof for $n$. A node with a smaller proof number is more promising to find a proof. Analogously, the disproof number $dn(n)$ for $n$ is the minimum number of leaf nodes to be disproven to find a disproof for $n$. For proven terminal node $n$, $pn(n) = 0$ and $dn(n) = \infty$, and $pn(n) = \infty$ and $dn(n) = 0$ for disproven terminal node $n$. At an internal node $n$ with a set of children $S(n)$, proving one child leads to a proof for an OR node $n$, while all children must be proven for a proof of an AND node $n$ (and vice versa for disproof). Hence, $pn(n)$ and $dn(n)$ are: For an internal OR node $n$, $pn(n) = \min_{s \in S(n)} pn(s)$ and $dn(n) = \sum_{s \in S(n)} dn(s)$. For an internal AND node $n$, $pn(n) = \sum_{s \in S(n)} pn(s)$ and $dn(n) = \min_{s \in S(n)} dn(s)$. Figure 2(left) illustrates an example of proof and disproof numbers, where numbers on the top and bottom inside each node indicate proof and disproof numbers, respectively.

PNS [1] maintains proof and disproof numbers of each node in a search tree. Starting with the root until reaching a leaf node, PNS traverses down a tree in a best-first manner by selecting a child with the smallest proof number at each internal OR node, and a child with the smallest disproof number at each internal AND node (*selection*). PNS then expands the selected leaf node (*expansion*), and recalculates proof and disproof numbers of the nodes along the path back to the root (*backup*). For example, in Figure 2(right), PNS selects path $A \rightarrow C \rightarrow F$, generates two new leaf nodes $H$ and $I$, and updates proof and disproof numbers at $F$, $C$ and $A$. PNS repeats these steps until finding either a proof or disproof or exhausting its time/memory resources.

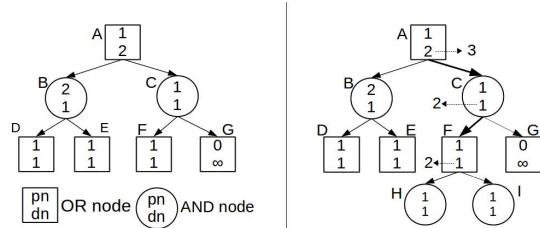

Figure 2: Example of PNS

Depth-First Proof-Number (DFPN) search [21] is a PNS variant that reformulates best-first PNS to depth-first search. DFPN has been successfully used to solve difficult games or game positions in many games, e.g., [2, 21, 27], often involving a solution with several hundred steps [13, 21].

DFPN re-expands fewer internal nodes than best-first PNS as well as operates in limited memory. A large part of the available memory is allocated to the transposition table (TT). The TT caches the proof and disproof numbers of examined nodes. DFPN introduces thresholds for proof and disproof numbers: $th_{pn}(n)$ and $th_{dn}(n)$ which are set to initially large values at the root node. DFPN recalculates $pn(n)$ and $dn(n)$ by using the proof and disproof numbers of $n$'s children. When $th_{pn}(n) \leq pn(n)$ or $th_{dn}(n) \leq dn(n)$ holds for an unproven node $n$, DFPN identifies that there are more promising nodes than $n$ and postpones the examination of $n$. Otherwise, DFPN selects a child $s_1$ with the smallest (dis)proof number for a further examination, with the following thresholds: For OR node $n$, $th_{pn}(s_1) = \min(th_{pn}(n), pn(s_2) + 1)$ and $th_{dn}(s_1) = th_{dn}(n) - dn(n) + dn(s_1)$. For AND node $n$, $th_{pn}(s_1) = th_{pn}(n) - pn(n) + pn(s_1)$ and $th_{dn}(s_1) = \min(th_{dn}(n), dn(s_2) + 1)$, where $s_2$ is a child with the second smallest (dis)proof number among a list of children of an OR (AND) node $n$. DFPN tends to gradually increment the thresholds as search progresses.

The basic PNS sets $pn(n) = dn(n) = 1$ for an unproven leaf $n$. *Heuristic Initialization* enhances PNS variants including DFPN+ [21] by initializing $pn(n) = h_{pn}(n)$ and $dn(n) = h_{dn}(n)$ at unproven leaf $n$, where $h_{pn}(n)$ and $h_{dn}(n)$ are evaluation functions for proof and disproof numbers. Existing work on creating $h_{pn}(n)$ and $h_{dn}(n)$ for games includes manual [16, 35] and machine learning [9, 21] approaches. Even if DFPN reduces re-expansions of internal nodes compared to basic PNS, it still often suffers from the high overhead of such a re-examination, resulting in examining only small portions of the new search spaces. To alleviate this issue, approaches to increase $th_{pn}(n)$ and $th_{pn}(n)$ have been developed [16, 21, 24].

### 3.2 Monte Carlo Tree Search

MCTS is based on a best-first search that repeats the selection, expansion, and backup steps. Most MCTS algorithms except [32] perform Monte Carlo samplings to calculate a heuristic value of a leaf node [8, 18]. A sampling at a leaf estimates a probability of a win by randomly selecting moves until it reaches a terminal position. For more accurate evaluations, samplings select moves with non-uniform probabilities, considering game board configurations [10, 31]. Additionally, MCTS can use evaluation functions for leaf-node evaluations with [31] or without [32] sampling.

One conceptual difference between MCTS and PNS is that MCTS aims to balance a trade-off between *exploration* and *exploitation*. A move with a high reward (e.g., high winning ratio) must be selected to have a higher chance to win a game (exploitation). On the other hand, a move with a low reward must also be selected to overcome an inaccuracy due to a small number of examinations of the move (exploration). A variety of formulas [10, 18, 34] have been presented to achieve the right balance.

Let $n$ be the current node, $A(n)$ be a set of legal moves for $n$, and $Q(n, a_i)$ be an accumulated reward for move $a_i$. To select the best move $a_{best}$ in RA, consider Segler et al.'s formula [29], originally presented in [26, 31]: $a_{best} = \underset{a_i \in A(n)}{\arg \max}(\frac{Q(n,a_i)}{N(n,a_i)} + cP(n,a_i)\frac{\sqrt{\sum_{a \in A(n)} N(n,a)}}{1+N(n,a_i)})$ where $N(n, a_i)$ is the number of visits to $a_i$ at $n$, $P(n, a_i)$ is a prior probability of move $a_i$ at $n$, and $c$ is a constant empirically preset. In RA, $P(n, a_i)$ corresponds to a probability that one step retrosynthesis with reaction rule $a_i$ is successfully applied to molecule $n$. A neural network is trained to estimate $P(n, a_i)$

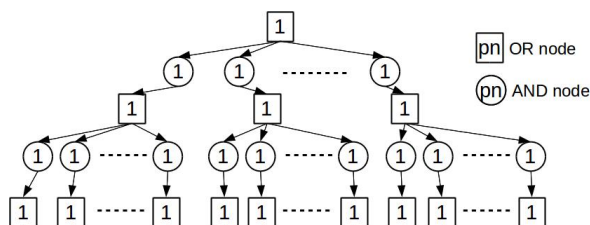

Figure 3: Proof number limitation in lopsided search space

[29, 30]. As in UCT [18], the first and second terms of the above formula take exploitation and exploration into account. However, the above second term more gradually reduces the exploration factor than UCT and encourages to explore the search space based on $P(s, a_i)$ encoding domain-specific knowledge.

Let $b_i$ be a sequence of selected moves in a search tree leading to node $n$ from the root, $L(b_i)$ be its length, and $n_j$ be a node with selected move $a_j$ on $b_i$. Segler et al. define $Q(n, a)$ to prefer a move with a higher winning ratio as well as a shorter pathway leading to a win: $Q(n, a) = \sum_{i=1}^{n} I_i(n, a) z_i \max(0, 1 - \frac{L(b_i) - \sum_{a_j \in b_i} k P(n_j, a_j)}{L_{max}})$ where $I_i(n, a)$ is a function indicating whether move $a$ at node $n$ is selected at the $i$-th traversal, $k$ is a damping constant ($0 \leq k \leq 1$), $L_{max}$ is the maximum tree depth, and $z_i$ is a reward received depending on whether a sampling leads to a win or not. A cut-off depth $d_r$ stops a sampling that reached no terminal node.

## 4 Proof-Number Search with Heuristic Edge Initialization

In finding a proof at the root, PNS tends to select a path that has nodes with small proof numbers. Since the proof numbers of OR children are summed up to calculate the proof number of an AND node, PNS prefers selecting an AND node that has a small number of moves. In games where PNS variants have been successfully applied, e.g., [16, 21, 27], the number of available moves can change significantly between consecutive game positions. Such a property of the opponent's move distribution in the game search space allows PNS variants to have various values of proof numbers, successfully identifying promising portions of the search space.

RA raises a new challenge due to different characteristics of its search space. In RA, many reaction rules are applicable to a molecule, increasing the number of moves at OR nodes. However, most of the reaction rules have only one precursor. There are at most a few reactants in the most complicated reaction rules. In other words, the search space is *lopsided*, since the branching factor at OR nodes is large, while it is very small at AND nodes. In the lopsided search space, PNS variants have difficulties in identifying moves with higher chances of leading to proofs. For example, assume that the AND nodes illustrated in Figure 3 always have only one move each. Then, even if there are many moves at OR nodes, $pn(n) = 1$ holds for any node $n$ in the figure, thus preventing PNS variants from identifying a most promising path leading to a proof. In this figure, DFPN is turned into simple, inefficient depth-first search with no depth limit.

Heuristic initialization at leaf nodes can assign a variety of proof numbers to leaf nodes and turns PNS variants into greedy search in the lopsided search space. This approach suffers from performance degradation when an evaluation function $h_{pn}(n)$ assigns an inaccurate proof number to a leaf. Assume that $h_{pn}(n)$ evaluates all leaf nodes illustrated in Figure 3, only the left leaf node $l$ has a proof, and $h_{pn}(l) > h_{pn}(m)$ holds for any other leaf $m$. Then, PNS needs to examine all the nodes in this figure, before leaf $l$ is examined.

Our DFPN with Heuristic Edge Initialization (DFPN-E) addresses a limitation arising in the lopsided search space. DFPN with heuristic initialization of *leaf* nodes attempts to sum up the values of the leaf nodes that can be part of a proof tree. Based on more informed heuristics values calculated by a set of the leaf nodes, DFPN with heuristic initialization determines a leaf to expand next. In contrast, DFPN-E assigns a heuristic cost to an *edge* from an OR node to an AND node, which estimates the difficulty of finding a proof. In addition to the number of leaf nodes needed to find a proof, DFPN-E attempts to estimate the total cost of the edges needed to be included in a proof tree.

A leaf node to expand next is determined by this estimated effort. Once DFPN-E proves that an edge can lead to a proof, it does not need any effort to find a proof. DFPN-E, therefore, updates the edge cost to zero. Formally, we define $pn(n)$ for an internal OR node $n$ as follows:

$$pn(n) = \begin{cases} 0 & (\min_{s \in S(n)} pn(s) = 0) \\ \min_{s \in S(n)}(h(n,s) + pn(s)) & (\text{otherwise}) \end{cases}$$

where $h(n,s)$ is an evaluation function evaluating an edge from node $n$ to its child $s$. In addition, the evaluation value is non-negative for any $n$ and $s$. The remaining cases of calculating $pn(n)$ and $dn(n)$ as well as the child selection scheme are the same as described in the previous section for PNS.

Heifets' and Jurisica's PNS implementation [11] is regarded as a special case of $h(n,s) = 1$ for any $n$ and $s$. In this case, their PNS implementation is almost identical to iterative deepening based on depth except that it always returns to the root whenever a leaf is expanded. While an advantage of PNS in solving games is to examine the search space as deep as possible without any depth limit, their approach no longer inherits this advantage, losing the capability of finding long pathways.

As in DFPN, DFPN-E uses two thresholds $th_{pn}(n)$ and $th_{dn}(n)$: one for the proof number and the other for the disproof number, enabling DFPN-E to search as long as $th_{pn}(n) > pn(n) \wedge th_{dn}(n) > dn(n)$ holds. Let $s_{best}$ be a child chosen for an examination at an OR node $n$. DFPN-E updates $th_{pn}(n)$ in a combination of [21] with [16]:

$$th_{pn}(s_{best}) = \min(th_{pn}(n), pn(s_2) + \delta) - h(n, s_{best})$$

where $s_2$ is a child where $pn(s_2)$ is the second smallest proof number with our modification among a list of $n$'s children, and $\delta$ is an integer for threshold controlling to decrease the node reexamination overhead [16]. The remaining cases are identical to original DFPN [21].

Our evaluation function $h(n,s)$ is based on a one-step retrosynthesis prediction combined with the idea behind so-called *forced moves* in games. First, as described in [30], our approach encodes a molecule into a *fingerprint* that is a fixed bit vector. A neural network, described in the next section, receives the fingerprint as input. The neural network has $R$ nodes in its output layer where $R$ is the number of reaction rules in the reaction database. The neural network predicts a probability $P(n, a)$ that reaction rule $a$ is applied to a molecule $n$ in a reverse manner. Then, we define $h(n, s)$ as follows:

$$h(n, s) = \min(M_{pn}, \lfloor -\log(P(n, a_i) + \epsilon) + 1 \rfloor)$$

with $M_{pn}$ a constant, $a_i$ the reaction rule to generate a child $s$ at OR node $n$, and $\epsilon$ a small constant.

Finally, let $a_{i-1}$ be the reaction rule applied just before $a_i$ in the current search tree. On top of the above $h(n, s)$, our approach sets $h(n, s) = 0$, if (1) $a_{i-1} = a_i$ holds and (2) the largest molecule in $s$ is smaller than that in $n$. We define the size of a molecule as the number of atoms. The intuition behind the latter heuristic rule is that a simpler precursor molecule could often be used to synthesize a product, as the feasibility of a simpler molecule tends be easier to verify. In addition, applying the same rule encourages DFPN-E to keep simplifying the molecule structure. Forced moves in games have similar principles to guide search towards positions that can be more easily verified.

## 5 Experimental Results

This section is dedicated to empirical evaluations of our algorithms.

### 5.1 Setup

Our implementation easily solves the benchmark instances in [11] with their reaction and starting material databases, except instance #18 that has a special symmetric structure. Therefore, for our empirical evaluation, we create more difficult benchmarks from reactions extracted from the text mining of US patents from 1976–2013 provided by Lowe[2] [20]. We first select unique reactions that RDKit[3] can correctly parse, and split them into a training set and a test set. The training set contains 681,866 reactions in patents filed before ones in the test set. We use the training set to create databases of reaction rules and starting materials, and train a neural network.

We automatically construct reaction rules, as commonly done in the literature, e.g., [19, 29]. We keep only the largest molecule in the product of each reaction. From each reaction, we then extract a reaction core and its direct neighbors as a candidate for a reaction rule. If that candidate appears at least 8 times in the training set, we keep it. We obtain 1550 reaction rules. Molecules present in the training set are included in the starting material database, which ends up having 977,435 molecules. The training set of Segler et al. [29] is more than 18 times larger than ours (12.4 million version 681 thousand), because we do not have any access to commercial databases. Therefore, as a trade-off, we have a smaller number of reaction rules as well as use fingerprints and neural networks of a smaller size. Our fingerprints are based on the Morgan fingerprints [25] of radius 3 with 512 bits.

## 5.2 Implementation Details

The neural network for edge initialization consists of one fully connected layer of 512 neurons with Rectified Linear Unit (ReLU) activations followed by a softmax layer for 1550 output categories implemented in Tensorflow 1.12. The network parameters are optimized using Adam optimization with a learning rate of $10^{-4}$, batch size of 128 and a dropout rate of 0.2 applied after the dense layer.

From the test set, we extract 1945 target molecules that are not in the starting material database. We run each algorithm with a time limit of 900 seconds per instance on a machine whose CPU is Intel Xeon E5-2683 at 2.00GHz with 32GB memory and with only one core in use. Algorithms evaluated include: (1) **MCTS**: MCTS with Segler et al.'s formula [29], (2) **DFPN**: Basic DFPN, with no edge costs, (3) **DFPN$_H$**: DFPN with heuristic initialization of proof numbers at leaf nodes, (4) **DFPN-1**: DFPN with a unit edge cost, an enhanced version of Heifets' and Jurisica's PNS for chemical synthesis planning [11], and (5) **DFPN-E**: DFPN with heuristic edge initialization. For DFPN-E, we set $M_{pn} = 20$, $\epsilon = 10^{-30}$ and a threshold controlling parameter $\delta = 2$. For MCTS, we use parameters shown in [29], i.e., $c = 3$, $k = 0.99$, $L_{max} = 25$, $d_r = 5$, $z = 10$ if leading to a win and otherwise $z = -1$.

**Enhancements to MCTS.**    Unlike [29], our MCTS implementation uses only one neural network and performs neither forward pruning nor in-scope filtering, because we have a smaller set of reaction rules extracted from a smaller training set and because the purpose of our paper is to elucidate the performance of MCTS and DFPN under the game-solving model presented in [11]. However, we further enhance MCTS of Segler et al. [29] by formulating their Markov Decision Processes (MDPs) in a game-proof search. Assume that two reactants $B$ and $C$ are generated from a product $A$ by applying a reaction rule in a retrosynthetic manner, a state always includes both $B$ and $C$ in their representation. In contrast, our MCTS implementation regards it as an AND node with two OR children $B$ and $C$. This is more generic than the partial reward of Segler et al. [29], because $B$ are $C$ are considered as their *partial* state. In addition, at an AND node our MCTS implementation more dynamically selects $B$ and $C$, based on which one is more promising to prove.

## 5.3 Results

Of 1550 reaction rules, there are 1213 rules that have only one reactant, 330 rules with two reactants and 7 rules with three reactants, demonstrating that the search space is lopsided in RA. 78% of the rules generate only one move at an AND node, and at most three moves can be generated there. We do not intend to argue that there are more uni-molecular reactions than bi-molecular reactions, but these statistics result from writing the reaction templates obtained from the patent literature in the reaction direction of large molecules *reacting* into smaller molecules to facilitate the search from a usually larger target molecule towards the usually smaller starting materials.

Table 1 summarizes the performance of all algorithms. We include the search performance for the instances where the best known pathway lengths are longer than two, since we are interested in solving difficult instances. However, *there are two instances with a pathway length of two which are not solved by MCTS but solved by DFPN-E*. This shows that even finding extremely short pathways can fail if MCTS continues examining the wrong branches due to slow node expansion rates and a large branching factor. There are 897 instances in total, and 483 instances are solved by all methods with a time limit of 15 minutes per instance. The runtime and node expansion in Table 1 indicates the total runtime and the total node expansion that each algorithm needs to solve these 483 instances.

Table 1: Summary of performance

| Method | MCTS | DFPN | $DFPN_H$ | DFPN-1 | DFPN-E |
|---|---|---|---|---|---|
| Runtime (s) | 18,552 | 20,133 | 30,822 | 46,542 | **5,654** |
| Node expansion | 184,347 | 730,241 | 668,421 | 460,738 | **68,719** |
| Num solved | **852** | 770 | 619 | 691 | 842 |
| Longest pathway | 21 | 2,000 | 455 | **8** | 19 |
| Average pathway | 5.58 | 227.42 | 21.09 | **3.59** | 5.72 |

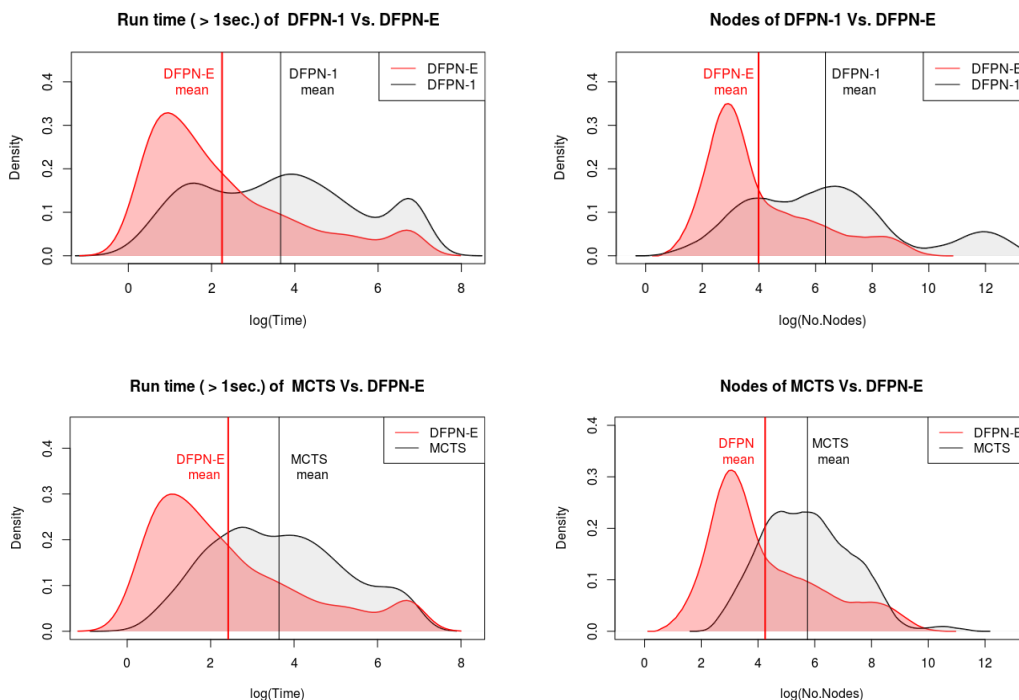

Figure 4: Performance comparison based on density distribution on instances.

DFPN-E tends to solve instances much more quickly than the other algorithms by significantly decreasing node expansions. For example, on average DFPN-E is 3.3 times faster than MCTS and reduces node expansions by a factor of 2.7. Compared to DFPN, DFPN-E improves the runtime by a factor of 3.6. We plot the logarithmic value of the runtime/node expansion on the vertical axis against the density distribution on the instances on the horizontal axis in Figure 4. This figure also clearly shows that DFPN-E performs better than DFPN and MCTS. $DFPN_H$, which initializes proof numbers at leaf nodes, worsens the performance of DFPN even if DFPN-E's evaluation function is used, demonstrating the importance of assigning heuristic values not to leaf nodes but to edges.

In terms of the number of problems solved, MCTS and DFPN-E are two of the most competitive algorithms. MCTS performs slightly better than DFPN-E. Of the 897 instances, both DFPN-E and MCTS solve 809 instances. DFPN-E solves 33 instances unsolved by MCTS, while there are 43 instances solved only by MCTS. The remaining 12 problems remain unsolved.

Both DFPN and $DFPN_H$ tend to return very long pathways. DFPN tends to search as deep as possible if an AND node has one branch. The lopsided search space of RA leads DFPN to returning a pathway of 2000 steps. In this case, DFPN keeps making very small changes to the target molecule such as changing one double bond to a single bond until it reaches a set of simple starting materials. DFPN's performance is degraded, especially when it accidentally fails to select a correct move immediately leading to a proof. In addition, synthesizing a target molecule with a long pathway is difficult for the chemist to perform in practice. Even with an evaluation function, $DFPN_H$ has a pathway of

455 steps in the worst case, showing that DFPN$_H$'s behavior is also affected by the lopsided search space. DFPN-1 returns the shortest pathways on average. However, as DFPN-1 performs similarly to depth-first iterative deepening limited by its search depth, it does not find pathways longer than 8. DFPN-1 is 8.2 times slower on average and solves 151 fewer instances than DFPN-E. We conclude that DFPN-E and MCTS are the strongest candidates for chemical synthesis planning.

## 6    Related Work

Nagai also introduces edge costs in DFPN [21]. His approach, DFPN+, initializes proof and disproof numbers at the *leaf* nodes and uses a *uniform* edge cost (e.g., -1 in the game of Othello [22]). The edge cost of DFPN+ aims to control the re-examination overhead of DFPN. In contrast, DFPN-E assigns *non-uniform* edge costs whose values depend on the promise of each edge, and is designed to work in lopsided search spaces. DFPN-E deals with the re-examination overhead with the formula of Kishimoto and Müller [16].

Optimal AND/OR search algorithms [15, 23] have constant edge costs and an admissible heuristic function to evaluate leaf nodes and calculate so-called q-values backed up in a manner similar to proof numbers. These approaches do not heuristically initialize edge costs. In addition, they monotonically increase the q-values to ensure optimality. However, all PNS variants including DFPN-E have a scenario where updated proof-numbers are decreased as search progresses, when one of OR children is proven at an AND node. When this scenario occurs, the search space rooted at that node becomes more promising to find a proof, which is a key advantage for DFPN-E.

There are other approaches based on OR search to perform retrosynthetic analysis, e.g., [19, 33]. Segler et al. [29] compare MCTS with greedy best-first search (GBFS) based on OR search and conclude that MCTS outperforms GBFS. Schreck et al. [28] employ deep reinforcement learning with a value network trained by using simulated experiences. Whether or not the policy learned by their approach could be effectively combined with DFPN-E remains important future work.

## 7    Conclusions

We elucidated the behavior of MCTS and PNS in the search space of RA and presented DFPN-E, a DFPN-based algorithm with edge cost initialization that addresses an essential problem of the lopsided search space in RA. Our experiments in search spaces generated from the US patent literature demonstrate that DFPN-E is a competitive alternative to MCTS for chemical synthesis planning. In future work, we plan to further improve the search performance of DFPN-E by incorporating successful approaches from other games as well as more accurate neural networks in cases of less data than in [29]. Furthermore, combining MCTS and DFPN-E in a portfolio and considering the feasibility of reactions [29, 33] are promising research ideas.

## Footnotes

*Work performed while the author was affiliated with IBM Research, Ireland.

[2]https://bitbucket.org/dan2097/patent-reaction-extraction/downloads

[3]https://www.rdkit.org/

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
