[Supplementary Material]

# Supplementary Material: Depth-First Proof-Number Search with Heuristic Edge Cost and Application to Chemical Synthesis Planning

## 1 Example of DFPN

When basic PNS updates a tree after generating nodes $H$ and $I$ illustrated in Figure 1 (right), $pn(A)$, $pn(C)$, $pn(F)$ remain unchanged. PNS traverses path $A \to C \to F$ again when it attempts to find a leaf node to expand next, starting from $A$.

DFPN addresses re-expansions of nodes $A$, $C$ and $F$ by introducing the thresholds of the proof and disproof numbers $th_{pn}(n)$ and $th_{dn}(n)$ [2]. DFPN keeps examining the search space rooted at node $n$ as long as it holds that $pn(n) < th_{pn}(n) \wedge dn(n) < th_{dn}(n)$.

In Figure 2, we illustrate an example of updating $th_{pn}(n)$. For the sake of simplicity, we deal with only $th_{pn}(n)$, since $th_{dn}(n)$ is updated in an analogous way.

DFPN's $th_{pn}(n)$ is used to decide whether DFPN needs to select a different path than the current one. In Figure 2(left), DFPN starts with $th_{pn}(A) = $ MAXVAL, which is a large value indicating that DFPN keeps examining the search space rooted at $A$ until $A$ is solved. $C$ is currently the best child of $A$ to examine because $pn(C) = 1 < pn(B) = 2$. On the other hand, when $pn(C)$ increases and $pn(B) < pn(C)$ holds, $B$ will become the best child to examine. To capture this, DFPN sets $th_{pn}(C) = 3$. This indicates that, as long as $pn(C) < 3$, $C$ remains the best child.

At node $C$, DFPN calculates $pn(C) = pn(F) + pn(G) = 1 < th_{pn}(C) = 3$. DFPN still examines the search space rooted at $C$, and selects $F$ because $dn(F) < dn(G)$ ($G$ is already proven). If $pn(C) = pn(F) + pn(G) \geq th_{pn}(C)$ holds, DFPN must select $B$. That is, when $pn(F) \geq th_{pn}(C) - pn(G)$ holds, DFPN must move to $B$, due to the fact that $pn(B) < pn(C)$. Therefore, DFPN sets $th_{pn}(F) = th_{pn}(C) - pn(G) = 3$.

Since $pn(F) = \min(pn(H), pn(I)) = 1 < th_{pn}(F) = 3$, DFPN can continue exploring the current path. Since $pn(H) = pn(I) = 1$, selecting either $H$ or $I$ looks equally promising. Assume that $H$ is chosen for an examination. Since $I$ becomes the best child when $pn(H) > pn(I) = 1$ holds, DFPN sets $th_{pn}(H) = 2$.

As in Figure 2(right), DFPN expands $H$ and recalculates $pn(H) = pn(J) + pn(K) + pn(L) = 3 > th_{pn}(H) = 2$. Therefore, DFPN updates $pn(H) = 3$ and recalculates $pn(F) = \min(pn(H), pn(I)) = 1 < th_{pn}(F)$. DFPN selects $I$, which is the best child of $F$, and does not propagate the proof and disproof numbers back to $A$. DFPN sets $th_{pn}(I) = \min(th_{pn}(F), pn(H) + 1) = 3$, indicating that path $A \to B$ becomes the best path when $pn(I) \geq th_{pn}(I)$ holds.

To enable DFPN to examine search as illustrated here, DFPN selects a child $s_1$ with the smallest (dis)proof number for a further examination, with the following thresholds:

- For OR node $n$, $th_{pn}(s_1) = \min(th_{pn}(n), pn(s_2) + 1)$, and $th_{dn}(s_1) = th_{dn}(n) - dn(n) + dn(s_1)$.

Figure 1: Example of PNS, adapted from the main paper

Figure 2: Example of DFPN

- For AND node $n$, $th_{pn}(s_1) = th_{pn}(n) - pn(n) + pn(s_1)$, and $th_{dn}(s_1) = \min(th_{dn}(n), dn(s_2) + 1)$.

where $s_2$ be a child with the second smallest (dis)proof number among a list of children of an OR (AND) node $n$. DFPN sets $pn(s_2)$ and $dn(s_2)$ to $\infty$ if node $n$ has only one child.

## 2 Pseudo Code of DFPN-E

Algorithms 1-2 show the pseudo code of DFPN-E. The essential differences from Nagai's DFPN and from DFPN+ [2] are highlighted in blue bold (shown in bold in gray scale print). DFPN-E uses a heuristic function $h(n, s)$ rather than a constant edge cost, additionally combined with a threshold controlling parameter $\delta$ of Kishimoto and Müller [1] except that $\delta$ is set to a constant in our DFPN-E implementation. The proof and disproof numbers of DFPN+ at leaf nodes are initialized by two evaluation functions $h_{pn}(n)$ and $h_{dn}(n)$, while DFPN-E currently sets $pn(n) = dn(n) = 1$ for the leaf nodes.

MAXVAL stands for a large integer. $S(n)$ is a set of children of node $n$. Node $n$ has 4-fields in addition to a state: a threshold for the proof number $thpn$, a threshold for the disproof number $thdn$, a proof number $pn$ and a disproof number $dn$. $TT$ is a transposition table that has fields to store a proof number $pn$ and a disproof number $dn$ in each transposition table entry. The hash key of node $n$ is calculated by the Zobrist function [3], which is commonly used in the game research community.

The IsStartingMaterial method checks if a node $n$ is a molecule in the starting material database. The NoApplicableReactionRule method checks if a node $n$ has no reaction rules applicable to $n$. The GenerateChildren generates the children of $n$.

For the sake of simplicity, we omit more detailed, efficient pseudo code, such as finding $s_{best}$ and $s_2$ while calculating $pn(n)$ and $dn(n)$ in one single for-loop. This can be embodied without any difficulty, and our actual code implements it.

## References

[1] A. Kishimoto and M. Müller. Search versus knowledge for solving life and death problems in Go. In *AAAI*, pages 1374–1379, 2005.

**Algorithm 1** DFPN-E

**Require:** Root node $r$
1: $r.thpn = r.thdn = \text{MAXVAL}$
2: $pn = \text{Search}(r)$
3: **if** $(pn = 0)$ **then**
4:     **return** PROOF
5: **else if** $(pn = \infty)$ **then**
6:     **return** DISPROOF
7: **else**
8:     **return** UNKNOWN
9: **end if**

[2] A. Nagai. *Df-pn Algorithm for Searching AND/OR Trees and Its Applications*. PhD thesis, The University of Tokyo, 2002.

[3] A. L. Zobrist. A new hashing method with applications for game playing. Technical report, Department of Computer Science, University of Wisconsin, Madison, 1970. Reprinted in *International Computer Chess Association Journal*, 13(2):169-173, 1990.

---

**Algorithm 2** Search

---

**Require:** Node $n$

1: **if** (IsStartingMaterial($n$)) **then**
2:    $TT[n].pn = 0; TT[n].dn = \infty$ //Proven terminal node
3:    **return** 0
4: **else if** (NoApplicableReactionRule($n$)) **then**
5:    $TT[n].pn = \infty; TT[n].dn = 0$ //Disproven terminal node
6:    **return** $\infty$
7: **end if**
8: GenerateChildren($n$)
9: **if** ($n$ is an OR node) **then**
10:    **loop**
11:        $n.dn = \sum_{s \in S(n)} s.dn$ //Calculate $dn(n)$ for an internal OR node
12:        **if** ($n.dn = \infty$) **then**
13:            $n.pn = 0$ //Proven internal OR node
14:        **else**
15:            $\boldsymbol{n.pn = \min_{s \in S(n)}(h(n,s) + s.pn)}$ //Calculate $pn(n)$ with heuristic edge cost initialization
16:        **end if**
17:        $TT[n].pn = n.pn; TT[n].dn = n.dn$ //Store updated search result
18:        **if** ($n.thpn \le n.pn \vee n.thdn \le n.dn$) **then**
19:            **break**
20:        **end if**
21:        $\boldsymbol{s_{best} = \arg\min_{s \in S(n)}(h(n,s) + s.pn); s_2 = \arg\min_{s \in S(n) \setminus \{s_{best}\}}(h(n,s) + s.pn)}$
22:        $\boldsymbol{s_{best}.thpn = \min(n.thpn, s_2.pn + \delta) - h(n, s_{best});}$
23:        $s_{best}.thdn = n.thdn - n.dn + s_{best}.dn$
24:        Search($s_{best}$)
25:    **end loop**
26: **else**
27:    **loop**
28:        $n.pn = \sum_{s \in S(n)} s.pn$ //$n$ is an AND node
29:        $n.dn = \min_{s \in S(n)} s.dn$
30:        $TT[n].pn = n.pn; TT[n].dn = n.dn$ //Store updated search result
31:        **if** ($n.thpn \le n.pn \vee n.thdn \le n.dn$) **then**
32:            **break**
33:        **end if**
34:        $s_{best} = \arg\min_{s \in S(n)} s.dn; s_2 = \arg\min_{s \in S(n) \setminus \{s_{best}\}} s.dn$
35:        $s_{best}.thpn = n.thpn - n.pn + s_{best}.pn$
36:        $s_{best}.thdn = \min(n.thdn, s_2.dn + 1);$
37:        Search($s_{best}$)
38:    **end loop**
39: **end if**
40: **return** $n.pn$

---