[Reviews · NeurIPS 2019]

Reviewer 1



Originality: PNS and related algorithms have not been evaluated for synthesis planning since work by Heifets and others several years ago. Revisiting this class of algorithms and proposing modifications to improve performance in multi-step synthesis planning is nice to see. Quality: The empirical evaluation is not as strong as it could be, but the conceptual contribution of this work is still important for the problem of synthesis planning. Clarity: The description of algorithms in 254-266 and elsewhere is not complete enough to reimplement the models and baselines. The dataset split, details of template extraction, network training, etc. is not provided either and the code is not available. Significance: The novelty of the modifications to the algorithm may be minor, but evaluating it in the context of this problem is important. However, the empirical evaluation is not as informative as it would need to be to support the claims of DFPN-E and MCTS being the strongest candidates for chemical synthesis planning. Specific comments: - The sentence “However, as discussed by [14], their search performance in RA is not well understood” prompted me to follow that reference, which does not really describe this. - It would make more sense for Figure 1 to show a retrosynthetic reaction rule - Reference https://doi.org/10.1021/acscentsci.9b00055 provides a different search strategy and might belong in the related work - The comment on line 189 (“However, most of the reaction rules have only one precursor”) and statistics of the rule sets described on lines 267-269 is surprising – it’s unexpected for there to be more unimolecular reactions than bimolecular reactions. - On line 235, was the data randomly split? - Line 241-242: Considering molecules in the training set as being starting materials seems overly generous. The molecules in the test set will be very similar to the training set if a random split was used, so the necessary pathways are quite short (as confirmed by the average pathway length in Table 1 – I suspect that a few outliers bring this average up). The histograms of the number of nodes shows a massive number require a single node. Given that the emphasis of this work is on multi-step synthetic planning, it would be better to exclude such simple test cases. --- Based on the authors' response (intent to release code, acknowledgement of reproducibility issue, and additional evaluation focusing targets with pathway lengths >= 3), a revised version would likely raise my score above the acceptance threshold

Reviewer 2



This reviewer really enjoyed reading this paper. PNS is a very interesting search algorithm which fell a bit behind MCTS after the introduction of progressive bias/PUCT. The difficulty of combining PNS with heuristics is a problem that also this reviewer worked on without much success, which is why this reviewer particularly appreciates this contribution. The manuscript is overall well written, and well motivated, however, a few question remain, which should be addressed before accepting the paper. Quality: In their MCTS baseline, did the authors phrase it as in Winands et al. "MCTS Solver", and Segler et al. Nature 2018, where a high or even infinite reward is given for proved states (solved and within tree) and a standard reward of 1 is given for states which are solved in the rollout? Did they consider giving partial rewards for partially solved states which is hinted at but not very well described in this 2018 Nature paper? This will likely accelerate MCTS convergence. Along these lines, do the authors stop PNS when a proof has been found, or do they continue to find more/better solutions? Is search speed the most important criterion, or are there other criteria which can be more important? How easy is it to incorporate other costs into PNS, e.g. cost of starting materials?

Reviewer 3



This paper proposes a variant of DFPN to solve the lopsided search space by applying simple heuristics. DFPN-E estimates the difficulty of finding a proof and utilizes it as the edge cost from the OR node to the AND node. Besides, two thresholds are used as constraints to the size of the search tree. The paper is well-written, especially the comprehensive overview of retrosynthetic analysis. The logic is clear and notations are clean. Nice work. Generally, I think the story is well-motivated and conclusions are well supported by the empirical experiments, but the contribution is a little incremental and engineering. It may not attract that many audience in the ML community, and personally, I would expect more explorations on the algorithm, not just two simple heuristics. Since the current model can reach the state-of-the-art performance, it would make a more interesting story by considering the more advanced strategies (like mentioned in the Conclusion Section). Other comments: (1) In line 72, should use “lose” instead of “loss”. (similarly to the remaining ones)

[Author Response · NeurIPS 2019]

| Method | MCTS | DFPN | DFPN$_H$ | DFPN-1 | DFPN-E |
|---|---|---|---|---|---|
| Runtime (s) | 18,552 | 20,133 | 30,822 | 46,542 | **5,654** |
| Node expansion | 184,347 | 730,241 | 668,421 | 460,738 | **68,719** |
| Num solved | **852** | 770 | 619 | 691 | 842 |
| Longest pathway | 21 | 2,000 | 455 | **8** | 19 |
| Average pathway | 5.58 | 227.42 | 21.09 | **3.59** | 5.72 |

Thank you for your constructive comments. Please find our response below.

**To reviewer 1.** We are in the process of releasing our source code, but are still waiting for internal approval. We agree on the missing details about our algorithm in lines 254-266 and will add the missing details in the revised version. Regarding our enhancements to MCTS of Segler et al.: assuming two reactants $B$ and $C$ are generated from a product $A$ by applying a reaction rule in a retrosynthetic manner, their state always includes both $B$ and $C$. In contrast, our MCTS regards it as an AND node with two OR children $B$ and $C$. That way the choice of $B$ and $C$ is more dynamic, and based on which one is more promising to prove. The remaining parts of the algorithms (e.g., template extraction) are the same standard algorithms. The data is split based on the date of filing as described in line 236, "The training set contains 681,866 reactions in patents filed before ones in the test set."

The table and figures int this rebuttal present the search performance for the instances with best known pathway lengths $\geq 3$. These results are confirming claims made based on Table 1 and Fig. 4 in the paper. There are 897 instances in total and 483 instances are solved by all methods. Furthermore, *there are 2 instances with a pathway length of 2 which are not solved by MCTS but solved by DFPN-E*, which shows that even finding extremely short pathways can fail if MCTS continues examining the wrong branches due to slow node expansion rates and a large branching factors. We will incorporate the specific change requests in the revised version.

In the paper we did not intend to say that there are more uni-molecular reactions than bi-molecular reactions and we are updating it in the revised version. These statistics result from writing the reaction templates in the reaction direction of large molecules "reacting" into smaller molecules to facilitate the search from a usually larger target molecule towards the usually smaller starting materials.

**To reviewer 2.** Once MCTS/DFPN prove that a subtree of a node is a win/loss, that subtree is no longer examined. For MCTS the partially solved states of MCTS by Segler et al. are prioritized and updated, based on their formula and *Heifets' and Jurisica's game solving model*. The game solving model already incorporates the idea of Segler et al. by regarding their state with multiple molecules as one AND node with multiple OR children. Each of the OR children accumulates its own reward. In addition, if a proof tree of a node is a sequence of moves and a MCTS sampling finds that sequence, the node is regarded as a win which no longer needs to be examined either. Furthermore, we evaluated standard rewards for MCTS, but did not include it in the paper because of space constraints and because it performed poorly by only solving 658 instances (table above) with longer pathways than the version we presented in the paper. There are still hard to solve instances that remain unsolved in our benchmark, therefore search speed is one of the next most important criteria to consider to evaluate performance.

**To reviewer 3.** As reviewer #2 pointed out, achieving better performance with a combination of evaluation function and PNS variants is not trivial. We plan to emphasize it in the revised version. Evaluation functions for DFPN need to be called in a different way than for PNS variants. Both MCTS and DFPN-E can become a choice in chemical synthesis planning and both are important contributions. At present, MCTS has been much more studied than the PNS variants. Our results will allow other researchers to consider both MCTS and PNS as strong candidates in domains with lopsided search spaces like chemical synthesis planning. Without our contributions, they might refrain from investigating PNS, since the standard way of using an evaluation function for PNS results in poor performance, as we demonstrate with this paper.



[Meta-Review · NeurIPS 2019]

This paper is a genuine borderline case. The work presents an algorithmic advance on an application and provides empirical evidence of the improved performance. Many aspects of this are very well done: the problem is well explained and well motivated; the prior work is well explained and placed in a proper context; the conclusions are measured and humble. The work is solid, but in fact it could be more solid; the results could be improved in some significant and plain-enough ways. In short, the paper provides significant evidence of an algorithmic advance, but not the best evidence. The new algorithm also only ties the previous best algorithm rather than beating it. So we have real science, but not the most exciting science nor the gold standard of evaluation. I would have thought that such a thing would be appropriate for a timely conference publication, but perhaps NeurIPS is now more archival than that. The view expressed above is based on the three reviews, a bit of discussion after the author response, and my own complete reading through of the paper (after which I was independently tending toward accept). The three reviews are not greatly different from each other in their views of the paper. The most negative review was borderline reject, because the work was thought to be somewhat incremental. The middle review was nominally borderline reject as well, but the reviewer indicated that they would cross over to accept for the expected revision. The most positive review was a clear accept, and that reviewer argued that the advance was more than incremental. However, this reviewer did not make a strong positive case overall for acceptance. So this paper is in fact a borderline case. After further consideration from the Program Chairs, given the AC's own leaning towards accepting, an Accept (Poster) recommendation was settled on. [This meta-review was reviewed and revised by the Program Chairs.]